# Transportation to the Slaughterhouse: Can Training Reduce the Stress Response in Horses?

**DOI:** 10.3390/vetsci12060547

**Published:** 2025-06-03

**Authors:** Francesca Dai, Marica Toson, Daniela Bertotto, Alessandro Dalla Costa, Eugenio Ugo Luigi Heinzl, Francesca Lega, Michela Minero, Barbara Padalino, Anna Lisa Stefani, Samuele Trestini, Federica Maietti, Gloria Zonta, Guido Di Martino

**Affiliations:** 1Independent Researcher, 20100 Milano, Italy; 2Istituto Zooprofilattico Sperimentale delle Venezie, 35020 Legnaro, Italy; 3Dipartimento di Biomedicina Comparata e Alimentazione, Università degli Studi di Padova, 35020 Legnaro, Italy; 4Dipartimento di Medicina Veterinaria e Scienze Animali, Università degli Studi di Milano, 26900 Lodi, Italy; 5Dipartimento di Science e Tecnologie Agro-Alimentari, Università di Bologna, 40126 Bologna, Italy; 6Faculty of Science and Engineering, Southern Cross University, East Lismore, NSW 2480, Australia; 7Department of Land, Environment, Agriculture and Forestry (TESAF), University of Padova, 35020 Legnaro, Italy

**Keywords:** slaughter horse, infrared thermography, transport, welfare, training, stress behavior, fecal cortisol

## Abstract

Road transportation is a significant stressor for horses, so the benefit of training procedures was tested in individuals reared for meat production, which had limited interactions with humans and little prior transport experience. This study evaluated the impact of self-loading techniques applied on the farm for six weeks on the subsequent stress responses during a one-hour pre-slaughter journey. Compared with the control group, trained horses were loaded faster and showed a lower frequency of stress behaviors during both transportation and unloading. Therefore, this study suggests that positive reinforcement training may be a useful preventive measure to protect horse welfare during transportation.

## 1. Introduction

In farm animals, transport-related stress can have a negative impact on animal welfare and can influence the meat quality [1,2,3,4], potentially reducing profits. Appropriate management measures can reduce transport-related stress in farm animals. These include the proper evaluation of animal-related factors (such as their species, breed, age, temperament, behavior, and health status [5,6]), adequate route planning [7], and appropriate handling during loading and unloading [6]. Among pre-transport measures, proper preparation for loading is considered a key factor. Repeated non-aversive handling and habituation to transport procedures have been reported to significantly reduce loading stress in cattle [8] and equids [9,10,11]. In Europe, around 6 million horses are bred yearly, and around 26.000 heads are transported for slaughter, raising citizens’ concerns [12]. According to EC Reg. 1/2005 [13], these animals can be transported for up to 8 h.

Compared to other species, horses can be more susceptible to a number of disorders following transport, in particular injuries and rhabdomyolysis [14]. In sport horses, habituation to transport procedures was found to reduce the incidence of transport-related behavioral problems and injuries [15]. The positive reinforcement-based training of loading [9,16] and self-loading techniques [15], focusing on voluntary entry into a form of transportation, was found to reduce the loading time and stress during loading. While in sport horses, several pieces of research have evaluated the effect of training horses to load on their stress (i.e., [9,15,17,18]), information is lacking for slaughter horses. In contrast to sport horses, horses kept for meat production have minimal interactions with humans and are generally transported to the slaughterhouse without prior training [19,20,21], which may have adverse effects on their welfare [22,23,24].

Behavioral and physiological evaluations are needed to assess transport-related stress in animals [25]. Stress-related behaviors observed during loading include pawing, turning away, pull-backs, and kicking [9,10,17,26]. Reported physiological modifications related to transport stress in horses are increases in their cortisol levels [4,27,28,29], glucose levels [29], circulating T3, T4, and fT4 levels [30,31], heart rate [30,32], β-endorphin levels [33], ACTH levels [33], core temperature [29], neutrophil–lymphocyte ratios [29], and packed cell volume [32]. Cortisol and its metabolites are important indicators of adrenal activity. The concentration of cortisol or its metabolites can be measured in various body fluids or excreta, and their variation can be indicative of subacute and chronic stress. Previous research demonstrated that a self-loading training technique reduced the loading time and increased forward locomotion toward the truck [10], also minimizing the necessity for human intervention during loading [34]. We hypothesized that training slaughter horses to self-load would mitigate their overall stress response related to transportation. The present work further investigated the effects of self-loading training on horses’ welfare by assessing their behavior and different indicators of acute (eye temperature) and subacute (fecal cortisol metabolites) stress during the transportation and unloading phases.

## 2. Materials and Methods

### 2.1. Farm and Animals

The study took place from April to October 2018. Thirty-two Spanish Breton horses of both sexes (18 intact males; 14 females) were randomly divided into two mixed-gender pens (pen 1 = 14 horses; pen 2 = 18 horses) upon arrival at a farm located in northeastern Italy. The animals were imported from Spain at 6 months of age and were aged 15 ± 2.79 months at the time of the experiment.

The pens had an indoor area with deep straw bedding (6 × 6 m^2^) and an outdoor area with a concrete floor (10 × 12.8 m^2^), which was available when the climatic conditions were favorable (i.e., not rainy or snowy). The average daily temperatures at the time of the study were 15–28 °C, with 60–85% RH. The horses were fed with a total mixed ration ad libitum and had free access to fresh water provided using an automatic drinker. The horses had limited interactions with the farmer in the form of daily visual checks and feeding by means of a truck. Physical contact with the horses was avoided. All the groups remained stable during the fattening period.

The farmer agreed to participate in this study voluntarily and signed a written consent form providing all the information about the research activities.

### 2.2. Training Protocol

The horses in pen one constituted the control group (CG; 6 intact males; 8 females); they were managed according to the on-farm routine, and no training was provided. The horses in pen two were categorized as the Training Group (TG; 12 intact males; 6 females); they were subjected to non-aversive training to self-load, applying target training using positive reinforcement. The training protocol used to train the TG has been described in detail by Dai et al. [10]. Briefly, the horses were first trained to follow the target (a yellow stick) in the loading lane connecting the outdoor area to the truck, while being reinforced with a handful of flaked corn. The target was progressively moved towards the truck. Following the target, the horses were led to the beginning of the ramp and then left free to load on their own initiative. On the truck, the horses were free to explore and eat the food located inside. The horses were trained in small groups (3–4 individuals) and were free to unload and regroup with the other horses from their respective group as they willed. Training sessions were performed three times a week, from 9:30 a.m. to 1:30 p.m. and from 2:30 p.m. to 4:30 p.m., for a total of 6 weeks of training. Therefore, we were unable to calculate how many minutes each horse was subjected to training for. The success criterion was defined as self-loading (i.e., the horse entered the trailer) three times in one week. All the horses met the success criterion. The training sessions were then replicated once a week until the transportation (i.e., for up to 3 months) to the slaughterhouse in order to maintain their memory.

### 2.3. Transports

According to the farm agenda, the horses were transported loose to the slaughterhouse in small groups (two to three horses) using the same truck (3.85 × 2.55 m^2^). Transportation took place in the afternoon (at around 4 pm) on 14 different days (mean group size: 2.36 ± 0.48 animals). The farmer, as a routine husbandry practice, reduced the number of animals equally in each pen. Therefore, the treatment groups were balanced in relation to the transportation. The farm manager conducted all the phases of transportation (loading, transportation, and unloading) following the routine procedures and with minimal handling of the small groups of horses (e.g., moving fences to allow the horses to enter the loading lane, the encouraging horses from behind with their voice, and waving a stick in the air behind the horses only when they refused to move). All the horses were transported to the same slaughterhouse, undertaking a journey of about 1 h (43 km) using the same route. They were slaughtered at around 5 a.m. the following morning. The journey included urban and suburban roads and no highways.

### 2.4. Behavioral Evaluation

The loading and unloading procedures were video-recorded using a digital video camera (Canon Legria HFR88, Canon Inc., Tokyo, Japan) controlled by the experimenter, and a GoPro Hero 5 (GoPro, Inc., San Mateo, California) mounted inside the van was used to record the horses during transportation. Due to technical issues (e.g., camera failures, insufficient light inside the van, the camera falling), only 16 horses (T = 8; C = 8) were recorded during transportation. The horses’ behavior at different transport phases was extracted from the video recordings by an animal welfare scientist experienced in equine behavior analysis who was blind to the treatment conditions, using the software Solomon Coder beta 17.03.22 and a focal animal sampling method. The duration of the different behaviors was recorded, and the percentage of time spent performing each behavior was subsequently calculated. The ethograms used for the analysis of transportation and unloading (modified from [32,35]) are shown in Table A1 and Table A2. The results from the analysis of the loading phase are published in Dai et al. [10].

### 2.5. Infrared Thermography

Infrared thermography was used as a non-invasive system to evaluate stress [36,37,38,39,40]. An infrared camera (NEC Avio TVS500; Nippon Avionics Co., Ltd., Tokyo, Japan) with a standard optic system was used to record the eye temperature (°C). Thermographic images were collected of unrestrained horses in the corridor leading to the truck. At the slaughterhouse, images were taken in the corridor leading to the pen. This choice guaranteed that all images were taken from the same distance (0.5 m) and the same angle (90°). Several images per horse were taken, and only images that were perfectly in focus were subsequently analyzed. The pre- and post-transport images for each horse were collected on the same day, before loading and after the arrival at the slaughterhouse. Moreover, an image of the same Lambert surface was taken before data collection to determine the radiance emission and to nullify the effect of surface reflections on the tested horses [41]. Grayess IRT Analyzer 6.0 software (Grayess Inc., Bradenton, FL, USA) was used to determine the caruncle temperature, measuring the maximum temperature (°C) within a circular area traced around the caruncle.

### 2.6. Fecal Sampling

To assess the baseline cortisol levels, fecal samples were collected at rest between 10:00 am and 1:00 pm. Each horse was sampled on two subsequent days; the average of the two samples was used as the baseline value. Within a minute from defecation, each sample was collected from the soil using a container and immediately placed in dry ice until delivery to the laboratory. A further fecal sample was collected after slaughtering the horses (this event occurred at around 12 h after the departure time). The fecal samples were collected directly from the rectum ampulla after killing the horses. The material was immediately stored in dry ice until delivery to the laboratory.

### 2.7. Laboratory Analyses

The levels of fecal cortisol metabolites (11β-Hydroxyetiocholanolone, 11-Oxoetiocholanolone) were determined in horse feces by liquid chromatography/tandem mass spectrometry after a Solid-Phase Extraction (SPE cartridges: Bond Elut PLEXA, polymeric, 6 mL, 200 mg) clean-up step and derivatization with hydroxylamine. The analytes were separated by rp-HPLC using an Acquity UPLC BEH column (10 mm × 2.1 mm, 1.7 µm) from Waters (Milford, MA, USA). Analysis was performed using an Acquity UPLC system (Waters, Milford, MA, USA) interfaced with a Quattro Premier XE triple quadrupole mass spectrometer with an ESI interface in the positive ionization mode (Micromass, Manchester, UK). The standard addition technique was used for quantification. The method was validated, and the following performance parameters were determined: the linearity, precision (based on laboratory reproducibility: 11β-Hydroxyetiocholanolone = 15%, 11-Oxoetiocholanolone = 23%), and detection limit (2.5 ng/g for both molecules).

### 2.8. Statistical Analysis

The data were analyzed using STATA 17.0. All the variables were non-normally distributed, as shown by the Shapiro–Francia test, so nonparametric tests were used. To compare the distribution of two independent samples (trained vs. control horses), the nonparametric Wilcoxon Mann–Whitney test was applied after the evaluation of the homoscedasticity hypothesis using a robust Levene’s test. A nonparametric Wilcoxon signed-rank test for two matched datasets was used to compare the data at the two sampling times (before and after transportation). The following behaviors during transportation, having an asymmetric distribution left-skewed towards zero, were treated as qualitative variables (presence vs. absence): head shaking, yawning, eating, displacing, attempting to bite, licking, chewing, and pawing. Fisher’s exact test and a two-sample test of proportions were performed for these qualitative variables. Pearson’s chi-squared test was performed to test the association between the group and sex. Confidence intervals for percentages were calculated using normal or binomial exact approximation, depending on the percentage value. The following behaviors were not analyzed due to a frequency of occurrence of zero during transportation: biting, eating, yawning, kicking, pawing, threatening body language, head tossing, displacing, putting their nose outside, whinnying, turn their head, lateral movements, walking, and interacting with their neighbors. As for unloading, we did not observe the following behaviors: trotting forward, galloping forward, remaining still, rearing, kicking, mounting, pawing, and urinating. The differences were considered significant at *p* < 0.05.

## 3. Results

### 3.1. Behavioral Evaluation

The results regarding the horses’ behavior during transportation and upon unloading are reported in Table A3 (quantitative variables). With regard to the qualitative variables measured during transportation, significant differences were found between C and T horses for head shaking (62% vs. 12%, *p* = 0.019), licking (37% vs. 0%, *p* = 0.027), and chewing (87% vs. 12%, *p* = 0.001).

The unloading times were 112.66 ± 71.04 sec and 169.62 ± 93.13 sec in T and C horses, respectively. We observed higher values in C horses than in T horses, but the statistical difference was not significant: *p* = 0.13. During unloading, T horses tended to walk forward more than the C group (*p* = 0.08).

No significant association was found between the group and sex (*p* = 0.126).

### 3.2. Fecal Sampling

Data on the levels of fecal cortisol metabolites (11β-Hydroxyetiocholanolone, 11-Oxoetiocholanolone) are given in Table A4. Significant differences were found in the levels of fecal cortisol after transportation compared to the baseline values. The values for 11β-Hydroxyetiocholanolone and 11-Oxoetiocholanolone were also combined and are presented as the total cortisol metabolites in Figure 1.

### 3.3. Infrared Thermography

Data on the eye temperatures are reported in Table A4. In both groups, the eye temperature significantly changed from before transportation to after unloading (*p* = 0.003 in T vs. *p* = 0.009 in C). A more relevant increase was found in T horses (median Δ: 2.10°) than in C horses (median Δ: 1.35°).

## 4. Discussion

To the author’s knowledge, this was the first study to test the effect of self-loading training in slaughter horses on stress reduction during pre-slaughter transportation. The data on the loading phase [10] demonstrated a shorter loading time and a higher frequency of forward locomotion toward the truck in trained horses than in untrained ones. In contrast, no differences in the frequencies of stress-related behaviors were found. However, specific behaviors (e.g., rearing, kicking, mounting) manifested nearly exclusively in the control group. The present study evaluated the behavioral responses during transportation and unloading. Specific behaviors (e.g., head shaking, licking, and chewing) were shown more frequently by horses in the control group than trained horses. These behaviors have previously been identified as indicators of stress in horses during transportation [18,35]. These results suggest that training focused on the loading phase not only reduced the loading time but may also have had an influence on the transportation phase, reducing some stress-related behaviors. These findings confirm that appropriate training could contribute to reducing the development of stress during transportation, thus improving horse welfare [14,18,42].

Unloading could be a challenging phase too, due to stressful factors such as the steepness and slipperiness of the ramp and the novel environment which the animals are required to enter [21]. In previous studies, horses have been observed freezing inside the vehicle or exhibiting flight responses [43]. In our study, no significant differences were found in the behavior during unloading between the two groups; however, a certain trend toward significance was revealed regarding forward locomotion, which was more frequent in T horses. Stress-related behaviors (remaining still, kicking, pawing, mounting, defecating) were demonstrated more frequently by the horses in the control groups, but this difference was not statistically significant. This result may be explained by considering that the horses had to unload in a new and potentially aversive environment: the lairage pens. The horses in our study had not faced such an environment before. Unknown noises, odors, and visual stimuli could have influenced their behavioral response to unloading.

Taken together, the results of the present study highlight the disturbance caused due to transportation, with few differences between the groups. In fact, despite being trained in loading procedures, many factors could have acted as stressors during traveling and unloading as well, such as unfamiliar staff, vehicle vibrations, noises, the driver’s skills, the temperature inside the van, and the amount of ventilation [18,21,25,44]. These findings suggest the need for training not only in loading procedures but also for traveling and unloading, as has previously been indicated for sport horses (see [18] for a review). Simulations of journeys allow horses to repeatedly be exposed to the various stimuli involved in transportation in a controlled manner, facilitating their habituation and, therefore, likely reducing their stress responses [42,45].

When a stressor is applied, endocrine responses related to the HPA axis are triggered to improve the animal’s fitness, so cortisol and its metabolites are detectable at higher levels than their baseline ones. The magnitude of this increase can provide information on the potential welfare impairment. Werner and Gallo [46] quantified the blood levels of cortisol in slaughter horses one hour before the journey to the slaughterhouse and one hour after each potentially stressful stage of transportation: immediately after loading, at the end of the journey, immediately after unloading, after lairage, in the stunning box before stunning, and during exsanguination. The highest values were shown at the end of the journey, while the lairage time produced a significant decrease in the cortisol concentration.

Schmidt and colleagues investigated the stress response of sport horses with no recent transportation experience to journeys of 1, 3.5, and 8 h, respectively [47]. Their salivary cortisol levels showed an immediate three-times increase in all the groups, while the cortisol metabolites (11-etiocholanolone) in their feces showed a significant increase one day after transportation, reflecting the intestinal passage time, but only for the 3.5 and 8 h journeys. In our study, we found an earlier increase in the cortisol metabolites (11β-Hydroxyetiocholanolone, 11-Oxoetiocholanolone) for a 1 h journey, suggesting a higher/faster stress response compared to that of sport horses. Moreover, the data demonstrate the possibility of using fecal cortisol metabolites to assess the stress response within a relatively short timeframe.

The physiological findings are also supported by the infrared thermography results. Thermography has previously been used as a non-invasive technique to assess the stress responses in several species [35,36,39,48,49]. Butterfield and colleagues [50] used infrared thermography to evaluate the transport-related stress in horses treated with a nutritional supplement, which was able to reduce their stress responses before being transported; this study found differences between the treated and untreated horses, suggesting the possibility of using infrared thermography to evaluate the transport-related stress. In our study, we observed a significant increase in the eye temperature after unloading in all the horses, indicating a stress response related to transportation, but we did not find a greater increase in the temperature in C horses compared to T horses, suggesting that training had a limited impact on the stress of unloading in a new environment.

The limitations of our study may include the recruitment of mixed-sex groups; the groups were established by the farmer upon the horses’ arrival, and we were not allowed to modify them for the study. An influence of sex on the stress response has previously been identified in horses [51,52]. While we did not find a significant effect of sex on the observed stress response, further research on the potential impact of sex on horses’ stress response, specifically to transport stress, is desirable.

Another possible limitation concerns the limitations of the use of fecal cortisol metabolites as a marker of transport stress for a journey of a short duration (1 h). The fecal samples were collected after slaughtering the horses, approximately 12 h after the departure time. The results could have been influenced not only by transport-related stress, but also by the stress experienced during the holding period in the pens at the slaughterhouse and the waiting time before being slaughtered. This method was chosen due to practical and ethical considerations: this approach is more animal-friendly than blood and saliva sampling, which requires restraining animals unfamiliar with humans. However, fecal sampling after slaughtering the horses provided less precise information regarding the exact stressor that caused the variations in their cortisol metabolite levels. Further research combining different markers and under longer journey durations will be needed.

From a practical perspective, while a 6-week training schedule is feasible in a research setting, it may not be integrable in a commercial farm routine, where human–animal interactions are kept at a minimum and the farmer is already engaged in many different tasks and may not have the necessary skills to implement a program based on target training. Therefore, it would be worth developing an effective training program with feasibility in mind to offer farmers a practical tool to improve their animals’ welfare. Also, investigating the impact of training before transportation on meat quality could enhance the proactivity of farmers.

## 5. Conclusions

Pre-slaughter journeys shorter than one hour are stressful for slaughter horses. Self-loading training, besides reducing the loading time and encouraging forward locomotion onto the truck, also reduced some stress-related behaviors during transportation. However, habituation to all the transport phases should be considered to improve the effectiveness of a training protocol in reducing transport-related stress. Moreover, the current duration and frequency of training could limit the feasibility of integrating the protocol into an on-farm routine. Future research should evaluate the effectiveness of shorter training protocols and determine how many training sessions are needed for slaughter horses to learn to self-load.

## Figures and Tables

**Figure 1 vetsci-12-00547-f001:**
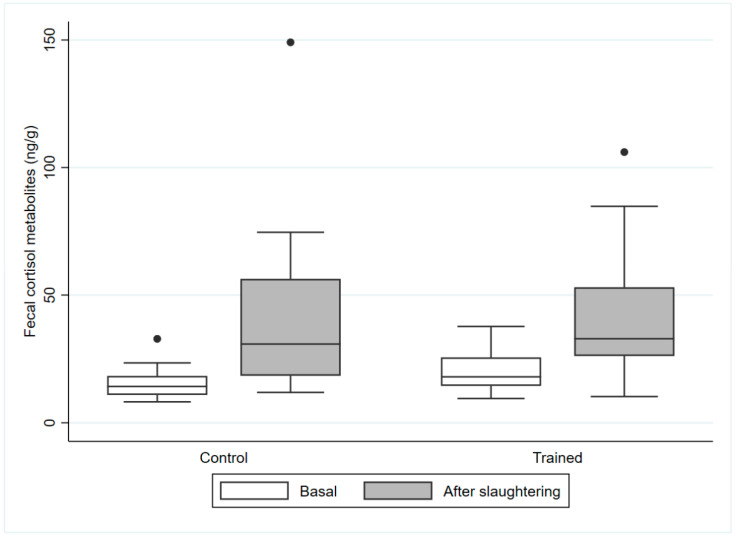
Boxplots showing the data distribution of fecal cortisol metabolites (11β-Hydroxyetiocholanolone and 11-Oxoetiocholanolone) at baseline and after transportation in two groups of slaughter horses which had (T) or had not (C) previously been trained to self-load. The band inside the box represents the median; the whiskers represent the lowest datum within the 1.5 interquartile range of the lower quartile and the highest datum within the 1.5 interquartile range of the upper quartile, while extreme outliers are indicated by ●.

## Data Availability

All the relevant data are published in this article.

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
