# Peer review of "Transportation to the Slaughterhouse: Can Training Reduce the Stress Response in Horses?"

_vetsci, 2025, doi:10.3390/vetsci12060547_

Round 1

Reviewer 1 Report

Comments and Suggestions for Authors

Dear authors,

Thank you for a well written and interesting manuscript. I only have minor comments.

Line 52-53: Is it 5.7 million horses pr. year?

Line 54: Which European countries does this take place in? I imagine it is more common in the south of Europe. Do you have any information regarding average transportation durations?

M&M section: When was this study carried out? 

Line 89: Delete "given".

Line 92: There is a blank space between of and daily - is a number missing?

Line 120: So some of the horses was transported alone - or together with horses not included in the study? If they were alone, what might the additional stress of being separated from the rest of the flock have done the cortisol levels?

Line 127: Is this rather short duration standard in Italy?

Line 254: Don't forget being handled by unfamiliar staff is also a stressor. The human factor is significant.

Line 275: Have you any thoughts on how the results could have been affected by longer transport durations in your study?

Line 295: You mention in the introduction that stress may affect meat quality. It would have been elegant had you included meat quality analyses in this study. Something to include in the next studies! Securing good meat quality could potentially motivate farmers to do a little extra work. 

Author Response

Thank you for a well written and interesting manuscript. I only have minor comments.

Authors: we thanks the reviewer for the nice comments.

Line 52-53: Is it 5.7 million horses pr. year?

Authors: addressed accordingly.

Line 54: Which European countries does this take place in? I imagine it is more common in the south of Europe. Do you have any information regarding average transportation durations?

Authors: we added information regarding the legal limits for transport duration in the EU.

M&M section: When was this study carried out? 

Authors: we included the year and the timeframe. The study took place from April to October 2018.

Line 89: Delete "given".

Authors: addressed accordingly.

Line 92: There is a blank space between of and daily - is a number missing?

Authors: addressed accordingly.

Line 120: So some of the horses was transported alone - or together with horses not included in the study? If they were alone, what might the additional stress of being separated from the rest of the flock have done the cortisol levels?

Authors: no horses were transported alone. The 14 transports included 2-3 study-horses each; we are sorry for the misunderstanding.

Line 127: Is this rather short duration standard in Italy?

Authors: unfortunately, we have no availability of average duration, which can be very variable. According to EU Regulation 1/2005 they can be transported up to 8 hours. We included the information in the introduction.

Line 254: Don't forget being handled by unfamiliar staff is also a stressor. The human factor is significant.

Authors: we added this factor as suggested.

Line 275: Have you any thoughts on how the results could have been affected by longer transport durations in your study?

Authors: Unfortunately, we were not able to carry out further studies with longer transports. We added a comment in the discussion recommending further research (lines 314-323).

Line 295: You mention in the introduction that stress may affect meat quality. It would have been elegant had you included meat quality analyses in this study. Something to include in the next studies! Securing good meat quality could potentially motivate farmers to do a little extra work. 

Authors: we added a comment on meat quality at lines 331-332.

Reviewer 2 Report

Comments and Suggestions for Authors

Manuscript ID vetsci-3609259: Transportation to the slaughterhouse

General comments

I recommend publication of the manuscript after revision.

The manuscript describes two groups of slaughter horses, one group that is trained to enter a truck and one control group that is not trained. The text is, at places, unclear. I have indicated some of the places below but I recommend that the text is also checked by an English speaking person.

Specific comments

Line 28 (and everywhere else in the manuscript): I think the term ‘meat horses’ is mostly used in French speaking countries and the term ‘slaughter horses’ is used in English.

Line 45: ‘could influence meat quality’. Given the small effect on behavior and the lack of effect on physiology the authors need to elaborate on this fact, possibly recommend inclusion of meat quality measurement in future studies.

Line 54-56: Compared to other species, horses can be more susceptible to a number of disorders following transport, such as pneumonia, diarrhea, colic, laminitis, injuries and rhabdomyolysis [13]. Yes, but these diseases hardly affect a horse that is going to be killed shortly after the transport.

Line 64: …and have adverse effects on their welfare [22–24]. Change sentence to ..something which may have adverse effects on their welfare [22–24].

Line 66 & 67: explain what planting and swings are.

Line 84: Change (intact males=18; females=14) to (18 intact males; 14 females).

Line 85-86: The sentence makes it sound like the horses agreed to participate on a voluntary basis.

Line 89-90: ‘given available when climatic conditions were favorable.’ I assume given available means that the horses were only released into the outdoor area “in good weather”. Please specify favorable (e.g. temp., wind, rain).

Line 93: The study is done on two groups of horses kept in two pens. One group (n = 14) is the control group, the other (n = 18) the trained group. The text needs to be written more clearly. E.g., when you write ‘Two pens were randomly selected to enter the study’, it sounds like there were other pens (or groups) not used for the study.

Line 95: Add The groups were…

Line 98: When you write ‘Horses in pen one were included in the Control Group’ it sounds like there were other horses included in the control group. Change to ‘The horses in pen one constituted the control group.’ or something like that. Change capital N to n = 14 and n = 18.

Line 102: Change In summery, to Briefly,

Line 107-110: The text is unclear. First you write ‘Horses were free to unload and regroup with the other horses from their respective group as willing’. Later you write ‘Horses were trained in small groups.’ Your description of how the horses were trained is unclear. (That’s the reason for my question in line 93.)

Line 125: Define ‘moving a stick’. Did he wave the stick in the air, did he touch the horses or did he sometimes hit them?

Line 136: Explain what you mean by ‘a treatment blind animal welfare scientist experienced in equine behavior analysis,’ I assume it was not him/her that was blind.

Comments on the Quality of English Language

See comments above

Author Response

The manuscript describes two groups of slaughter horses, one group that is trained to enter a truck and one control group that is not trained. The text is, at places, unclear. I have indicated some of the places below but I recommend that the text is also checked by an English speaking person.

Authors: we have revised English language as suggested.

Line 28 (and everywhere else in the manuscript): I think the term ‘meat horses’ is mostly used in French speaking countries and the term ‘slaughter horses’ is used in English.

Authors: we have reworded as suggested.

Line 45: ‘could influence meat quality’. Given the small effect on behavior and the lack of effect on physiology the authors need to elaborate on this fact, possibly recommend inclusion of meat quality measurement in future studies.

Authors: we have included this suggestion at the end of the discussion

Line 54-56: Compared to other species, horses can be more susceptible to a number of disorders following transport, such as pneumonia, diarrhea, colic, laminitis, injuries and rhabdomyolysis [13]. Yes, but these diseases hardly affect a horse that is going to be killed shortly after the transport.

Authors: we removed not pertinent conditions

Line 64: …and have adverse effects on their welfare [22–24]. Change sentence to ..something which may have adverse effects on their welfare [22–24].

Authors: addressed accordingly

Line 66 & 67: explain what planting and swings are.

Authors: the typo was amended (pawing). “Swing” has been defined as “turning head and body away from the ramp” (Waran & Cuddeford, 1995); the term has been substituted with “turn away”, which is clearer for the reader.

Line 84: Change (intact males=18; females=14) to (18 intact males; 14 females).

Authors: addressed accordingly

Line 85-86: The sentence makes it sound like the horses agreed to participate on a voluntary basis.

Authors: we rephrased the unclear sentence.

Line 89-90: ‘given available when climatic conditions were favorable.’ I assume given available means that the horses were only released into the outdoor area “in good weather”. Please specify favorable (e.g. temp., wind, rain).

Authors: the horses were released only in good weather, we have clarified this concept.

Line 93: The study is done on two groups of horses kept in two pens. One group (n = 14) is the control group, the other (n = 18) the trained group. The text needs to be written more clearly. E.g., when you write ‘Two pens were randomly selected to enter the study’, it sounds like there were other pens (or groups) not used for the study.

Authors: we rephrased this section.

Line 95: Add The groups were…

Authors: addressed accordingly

Line 98: When you write ‘Horses in pen one were included in the Control Group’ it sounds like there were other horses included in the control group. Change to ‘The horses in pen one constituted the control group.’ or something like that. Change capital N to n = 14 and n = 18.

Authors: addressed accordingly

Line 102: Change In summery, to Briefly,

Authors: addressed accordingly

Line 107-110: The text is unclear. First you write ‘Horses were free to unload and regroup with the other horses from their respective group as willing’. Later you write ‘Horses were trained in small groups.’ Your description of how the horses were trained is unclear. (That’s the reason for my question in line 93.)

Authors: we rephrased this section.

Line 125: Define ‘moving a stick’. Did he wave the stick in the air, did he touch the horses or did he sometimes hit them?

Authors: we clarified the sentence as suggested (waving a stick in the air behind the horses).

Line 136: Explain what you mean by ‘a treatment blind animal welfare scientist experienced in equine behavior analysis,’ I assume it was not him/her that was blind.

Authors: we clarified the sentence as suggested.

Reviewer 3 Report

Comments and Suggestions for Authors

Dear authors,

The present work evaluated the effects of self-loading training on the overall stress response to pre-slaughter transportation in meat horses. This approach derives from the need to find new strategies to mitigate the overall transport stress in this species. The topic is attractive because it provides useful information about the equids' stress coping during the transport in an observational physiologic research. However, these preliminary findings are not feasible and integrable in a commercial farm routine. So, the authors should better express in the discussion the practical applications of their results and the limits of their study. The references are specific and relevant. There are some typing errors in the text that should be fixed.  

Summary and Abstract: They are well written and summarize the information contained in the main text without repetitions. They provide background by placing the addressed question in a broader context, and include all the elements that should be present in an abstract and summary aligned with the editor's requirements.

Title, Keywords, and Introduction:  The title is attractive and adequate for the paper's content. I suggest you add some relevant words like: "Fecal cortisol" (instead of cortisol), "Stress behaviors" (instead of behavior), “infrared thermography”. The introduction is appropriate and effectively contextualizes the main aspects of the topic. The aim should be clearly mentioned and moved after the hypothesis, which is stated.

Materials and Methods: The study design is very simple but well-developed. It would be interesting to add information about the average environmental temperature and humidity of this period. All this information is important in a physiological and endocrinological study. Line 96: How many males and females were included in each group? I wouldn’t have mixed the sexes, especially if the males were stallions. There may be different sensitivities to stress between the sexes. This is a limitation of your study that should be discussed. It would be interesting to analyze the results based on sex, why did not you do this? Please add in the text the references from which you took the ethograms (they are present only in the captions). Paragraph of Fecal samples: Perhaps plasma or salivary cortisol should have been used to obtain a more accurate estimate for identifying the source of acute stress. Upon arrival at the slaughterhouse, multiple significant stressors are present, which may add to those experienced during transport. I believe this represents another limit of the study. Statistical analyses are well-performed and described; however, please move the information about the Software at the beginning of the paragraph. P value should be written in italic and lowercase letter and with the ≤ or ≥ symbol. Fix it in the text and caption where it appears.

Results, Discussion, and Conclusion: Results are logically presented and accompanied by clear tables and plots. The discussion, interesting and overall well done, follows a logical line and presents persuasive interpretations. The limits of the study are not expressed. Line 249-250: In addition to unloading, the holding period in the pens and the wait before slaughter may also have influenced the FCM results. It would have been more appropriate to use a different technique capable of assessing acute stress and applied immediately after unloading—an event already occurring in an adverse environment—to avoid the confounding impact of subsequent stressors. The limit of the study should be discussed. Lines 277-279: I agree with you, without doubt. However, it is also less precise in identifying what exactly was stressful for the animal. It might have been useful to perform both types of analyses—immediately after unloading and during the pre-slaughter holding period—to assess the progression of the animal’s stress response and to validate the use of FCM in this specific context. Please, discuss this study limitation and the practical application of your findings. The conclusion is adequate for the purpose.

Overall, the paper is well written and, since it adds useful information about the endocrinology of equids’ stress coping, it deserves to be published after minor revisions.

Author Response

The present work evaluated the effects of self-loading training on the overall stress response to pre-slaughter transportation in meat horses. This approach derives from the need to find new strategies to mitigate the overall transport stress in this species. The topic is attractive because it provides useful information about the equids' stress coping during the transport in an observational physiologic research. However, these preliminary findings are not feasible and integrable in a commercial farm routine. So, the authors should better express in the discussion the practical applications of their results and the limits of their study. The references are specific and relevant. There are some typing errors in the text that should be fixed.  

Authors: we have included this suggestion in the discussion

Summary and Abstract: They are well written and summarize the information contained in the main text without repetitions. They provide background by placing the addressed question in a broader context, and include all the elements that should be present in an abstract and summary aligned with the editor's requirements.

Authors: We thank the reviewer for the nice comments.

Title, Keywords, and Introduction:  The title is attractive and adequate for the paper's content. I suggest you add some relevant words like: "Fecal cortisol" (instead of cortisol), "Stress behaviors" (instead of behavior), “infrared thermography”. The introduction is appropriate and effectively contextualizes the main aspects of the topic. The aim should be clearly mentioned and moved after the hypothesis, which is stated.

Authors: addressed accordingly

Materials and Methods: The study design is very simple but well-developed. It would be interesting to add information about the average environmental temperature and humidity of this period. All this information is important in a physiological and endocrinological study. Line 96: How many males and females were included in each group? I wouldn’t have mixed the sexes, especially if the males were stallions. There may be different sensitivities to stress between the sexes. This is a limitation of your study that should be discussed.

Authors: we have included environmental data and the number of males and females in each group. The farm was organized in mixed gender pens and we had no possibility to change this choice. We have included a comment on this occurrence.

It would be interesting to analyze the results based on sex, why did not you do this?

Authors: the effect of gender was not significant, we added this information as suggested.

Please add in the text the references from which you took the ethograms (they are present only in the captions).

Authors: the references were added in the text (L147).

Paragraph of Fecal samples: Perhaps plasma or salivary cortisol should have been used to obtain a more accurate estimate for identifying the source of acute stress. Upon arrival at the slaughterhouse, multiple significant stressors are present, which may add to those experienced during transport. I believe this represents another limit of the study.

Authors: we included a comment on salivary cortisol in the discussion.

 Statistical analyses are well-performed and described; however, please move the information about the Software at the beginning of the paragraph. P value should be written in italic and lowercase letter and with the ≤ or ≥ symbol. Fix it in the text and caption where it appears.

Authors: addressed accordingly

Results, Discussion, and Conclusion: Results are logically presented and accompanied by clear tables and plots. The discussion, interesting and overall well done, follows a logical line and presents persuasive interpretations. The limits of the study are not expressed.

Authors: we included the limitations of the study in the discussion

Line 249-250: In addition to unloading, the holding period in the pens and the wait before slaughter may also have influenced the FCM results. It would have been more appropriate to use a different technique capable of assessing acute stress and applied immediately after unloading—an event already occurring in an adverse environment—to avoid the confounding impact of subsequent stressors. The limit of the study should be discussed.

Authors: we included a comment on salivary cortisol in the discussion

Lines 277-279: I agree with you, without doubt. However, it is also less precise in identifying what exactly was stressful for the animal. It might have been useful to perform both types of analyses—immediately after unloading and during the pre-slaughter holding period—to assess the progression of the animal’s stress response and to validate the use of FCM in this specific context. Please, discuss this study limitation and the practical application of your findings.

Authors: we included the limitations of the study in the discussion

The conclusion is adequate for the purpose.

Overall, the paper is well written and, since it adds useful information about the endocrinology of equids’ stress coping, it deserves to be published after minor revisions.

Authors: We thank the reviewer for the nice comments.

Round 2

Reviewer 2 Report

Comments and Suggestions for Authors

No comments